# Systematic review of the evidence on orthotic devices for the management of knee instability related to neuromuscular and central nervous system disorders

Catriona McDaid,[1] Debra Fayter,[2] Alison Booth,[1] Joanne O'Connor,[2] Rocio Rodriguez-Lopez,[3] Dorothy McCaughan,[1] Roy Bowers,[4] Cynthia P Iglesias,[1] Simon Lalor,[5] Rory J O'Connor,[6] Margaret Phillips,[7] Gita Ramdharry[8,9]

► Prepublication history and additional material is available. To view please visit the journal (http://dx.doi.org/ 10.1136/bmjopen-2017-015927).

For numbered affiliations see end of article.

**Correspondence to**
Dr Catriona McDaid;
catriona.mcdaid@york.ac.uk

## ABSTRACT

**Objectives** To assess the effectiveness of orthotic devices for the management of instability of the knee in adults with a neuromuscular disorder or central nervous system disorder.

**Design** A systematic review of primary studies.

**Setting** Community.

**Participants** Adults with a neuromuscular disorder or central nervous system disorder and impaired walking ability due to instability of the knee.

**Interventions** Orthoses with the clinical aim of controlling knee instability, for example, knee-ankle-foot orthoses, ankle-foot orthoses and knee orthoses or mixed design with no restrictions in design or material.

**Primary and secondary outcome measures** Condition-specific or generic patient-reported outcome measures assessing function, disability, independence, activities of daily living, quality of life or psychosocial outcomes; pain; walking ability; functional assessments; biomechanical analysis; adverse effects; usage; patient satisfaction and the acceptability of a device; and resource utilisation data.

**Results** Twenty-one studies including 478 patients were included. Orthotic devices were evaluated in patients with postpolio syndrome, poststroke syndrome, inclusion body myositis and spinal cord injury. The review included 2 randomised controlled trials (RCTs), 3 non-randomised controlled studies and 16 case series. Most were small, single-centre studies with only 6 of 21 following patients for 1 year or longer. They met between one and five of nine quality criteria and reported methods and results poorly. They mainly assessed outcomes related to gait analysis and energy consumption with limited use of standardised, validated, patient-reported outcome measures. There was an absence of evidence on outcomes of direct importance to patients such as reduction in pain and falls.

**Conclusions** There is a need for high-quality research, particularly RCTs, of orthotic devices for knee instability related to neuromuscular and central nervous system conditions. This research should address outcomes important to patients. There may also be value in developing a national registry.

### Strengths and limitations of this study

► The first systematic review addressing this question to systematically consider study quality.
► An extensive range of sources were searched to identify studies.
► It was difficult to be certain that knee instability was the main problem being treated in some of the studies.
► Due to poor reporting of the primary studies, it was not possible to extract outcome data in the standardised way planned in the protocol.

**Registration number systematic review** PROSPERO (CRD42014010180).

## INTRODUCTION

Instability can occur in any of the three anatomical planes of the knee: sagittal, coronal or transverse planes, and there are several mechanisms that may lead to knee instability in neuromuscular disorders (NMDs) and central nervous system (CNS) conditions. These include: weakness or over-activity of any of the muscles that have a direct effect on the knee (knee extensors, knee flexors and gastrocnemius) and muscle weakness or overactivity remote from the muscles directly affecting the knee due to secondary effects on posture (eg, alterations to the anterior progression of the ground reaction force under the foot or plantarflexor weakness leading to uncontrolled dorsiflexion). In the case of CNS conditions, spasticity in the muscles around the knee can also cause knee instability (eg, spasticity in the gastrocnemius causes excessive plantarflexion in stance that shifts the ground reaction force anterior to

the knee causing hyperextension).[1] Knee instability can lead to pain, falls and a range of mobility issues for the individual.

Knee instability due to muscle weakness or ligamentous laxity is often treated using orthoses with the functional goals of improving walking and to protect, stabilise and improve function.[2] Knee orthoses (KO) are often prescribed or in some cases a type of ankle-foot orthosis (AFO) known as a ground reaction AFO (GRAFO) may be provided. A GRAFO provides direct control of the ankle and foot, and indirect control of the knee and hip may be provided through optimising and normalising the alignment of the ground reaction force in relation to the knee joint throughout stance phase. A knee-ankle-foot orthosis (KAFO) is usually prescribed when bracing with an AFO or KO is insufficient to adequately control knee instability and usually when control in more than one plane is required.[2] Modern KAFOs tend to combine plastic and metal components: commonly polypropylene for calf and thigh shells and shoe inserts, aluminium, magnesium, titanium or steel for uprights and steel for joints.[3] Variations exist in the orthotic knee joint design, locking and unlocking mechanism, type of knee pads and plane of control.[3] A locked KAFO requires an altered gait to allow the individual's foot to clear the ground in the swing phase of walking. Polycentric knee joints can be locked or unlocked and permit a more anatomical or natural knee motion, though have more two-joint axes and may require more maintenance and are therefore more expensive.[3] Stance control knee joints have either a mechanical or microprocessor controlled knee joint that allows the knee to flex during the swing phase of walking, but locks during the stance phase of walking, when the knee is extended, allowing a more normal walking pattern. Other more extensive options include hip-knee-ankle-foot orthoses (HKAFO) originally designed for patients with higher level spinal cord dysfunction who might otherwise have been unable to walk.[4] These include hip guidance orthoses (HGOs) and reciprocating gait orthoses (RGOs), which have different locking mechanisms.

We undertook a systematic review with the aim of assessing the evidence base for the effectiveness of orthotic devices for management of instability of the knee in adults who have NMD or a CNS disorder. This was part of a larger mixed-methods project undertaken to inform the development of a future substantive research question on the clinical and cost-effectiveness of different types of orthotic management of the knee in people with NMD or CNS disorders.[1]

## METHODS
We undertook searches to identify studies assessing the effectiveness of orthotic devices for management of instability of the knee in adults who have NMD or a CNS disorder.

### Search methods for identification of studies
We searched the following databases from inception to November 2014: MEDLINE via Ovid, MEDLINE In-Process via Ovid, Cumulative Index to Nursing & Allied Health via EBSCO, EMBASE via Ovid, PASCAL via Ebsco, Scopus, Science Citation Index (ISI Web of Knowledge), BIOSIS Previews, PEDro, Recal Legacy, Cochrane Database of Systematic Reviews, Database of Abstracts of Reviews of Effects, Health Technology Assessment (HTA) database and Cochrane Central Register of Controlled Trials in The Cochrane Library, Conference Proceedings Citation Index-Science (ISI Web of Knowledge), Health Management Information Consortium via Ovid, ClinicalTrials.gov, WHO International Clinical Trials Registry Platform, National Technical Information Service and selected websites. There were no language or publication status restrictions. See online supplementary appendix 1 for the MEDLINE (Ovid) search strategy, which was adapted for the other databases.

The reference lists of all included studies, any related systematic reviews and key background papers were checked to identify any further relevant studies.

### Eligibility criteria
#### Population
Adults (16 years or older) with NMD or CNS disorder and impaired walking ability due to instability of the knee were eligible for inclusion. Children were excluded.

#### Intervention
Orthoses with the clinical aim of controlling knee instability, for example, KAFO, AFO and KO or of mixed design with no restrictions in design, material, custom or prefabricated; type of knee joint or stance-control design (KAFO), or whether there was an electronic component. Studies evaluating the use of functional electrical stimulation were excluded.

Studies were eligible provided the orthosis had been used in a real-life setting (ie, studies where the device had been solely used within a laboratory/experimental setting were excluded). Outcomes could be assessed in a laboratory or clinic setting provided participants had used the device in the community.

#### Comparator
Studies using any of the above orthoses as a comparator, including studies comparing different designs of the same orthosis or no intervention.

#### Study design
Randomised controlled trials (RCTs) and other study designs with and without a comparator group such as non-randomised controlled studies, before and after studies and case series were eligible for the review.

The following outcomes were of interest: condition-specific or generic patient-reported outcomes measures assessing function, disability, independence, activities of daily living, quality of life or psychosocial outcomes; pain; walking ability; functional assessments; biomechanical

analysis; adverse effects; usage; patient satisfaction and the acceptability of a device; and resource utilisation data.

Two researchers independently screened titles and abstracts and full papers to assess eligibility. Disagreements were resolved through discussion and consultation with a third member of the project team if necessary. Authors were contacted if eligibility was uncertain from the information provided in the publication. There were no language restrictions.

### Data extraction

Data were extracted by one researcher and checked by a second researcher with discrepancies resolved by discussion. Studies in languages other than English were extracted by a native speaker who was also a researcher and were checked by a second researcher for consistency only. Data were extracted using a piloted data extraction form. Multiple publications of the same study (linked papers) were extracted and reported as a single study. Between-group differences were extracted from studies with a comparator. We had planned to extract data to allow calculation of between group differences and CIs. However, due to the generally poor reporting of data, it was not possible to consistently do this across studies. Where data were available, these were extracted; where the appropriate data were not reported, the description of the results provided in the paper was extracted, and the lack of summary data was noted.

### Study quality

RCTs were assessed using the Cochrane risk of bias criteria.[5] Non-randomised studies with a control group were assessed for external validity, performance bias, detection bias and selection bias/control of confounding based on eight criteria (gender, age, cause of muscle weakness, presence of sensory disturbance, whether the orthosis was used for proximal or distal muscle weakness, previous use of an orthosis, acclimatisation time and type of orthosis used). Case series were assessed using criteria adapted from the assessment of controlled studies and criteria used in a previous systematic review.[6] Assessment of risk of bias was undertaken independently by two researchers (except for non-English language studies). Discrepancies were resolved by discussion.

The protocol was registered with PROSPERO in advance of undertaking the review (registration number CRD42014010180). Ethical approval was not required.

## RESULTS

### Overview of the evidence

A total of 4516 references were identified from the searches, and 21 studies of 478 patients (reported in 25 publications) were included (figure 1). A full list of papers and reasons for exclusion is available from the authors. A substantial proportion were excluded (n=76)

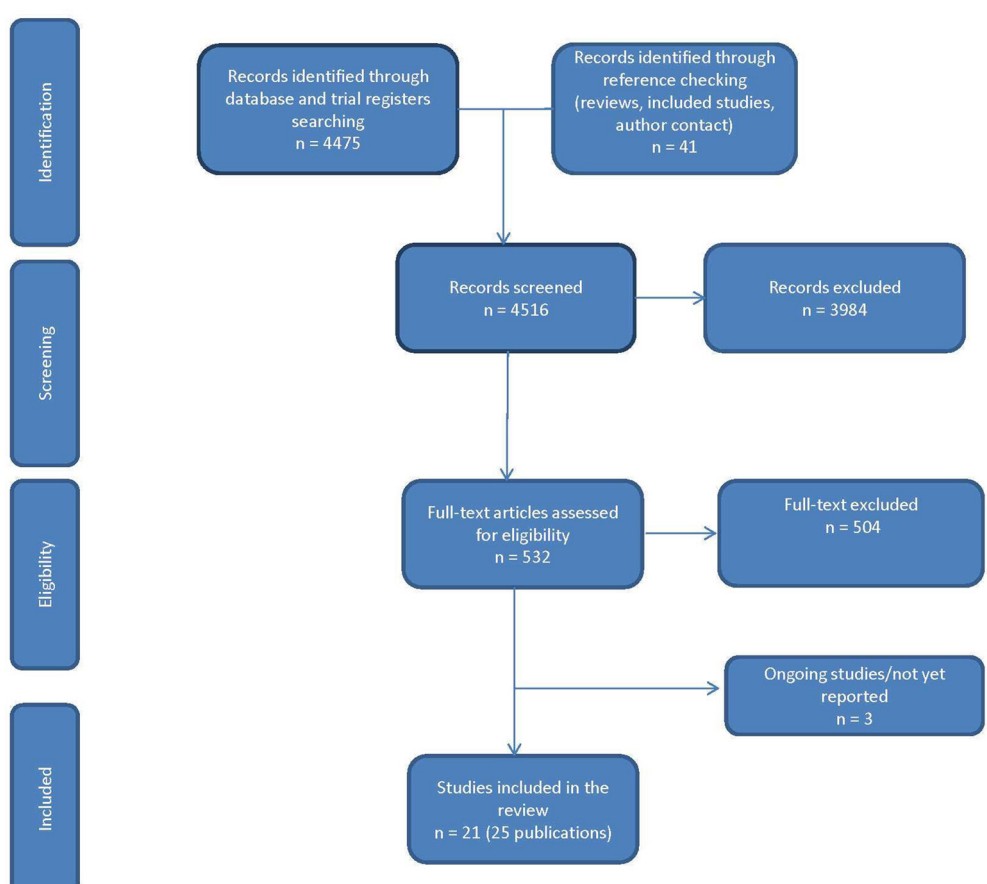

**Figure 1** Study selection.

because the orthosis was evaluated in a laboratory or clinical setting without the participant using the device in the community. Three potentially relevant ongoing studies were identified: a before and after study,[7] a case series[8] and an RCT.[9]

Table 1 provides a summary of study characteristics grouped by the four conditions covered by the included studies: postpolio, inclusion body myositis, poststroke and spinal cord injury. Two RCTs, 3 non-randomised studies with a control group and 16 case series were included. Sample sizes ranged from 5 to 67 participants and just fewer than half the studies had over 20 participants. The follow-up time was generally short, only six studies followed patients for 1 year or longer.

### Study quality
Both RCTs were assessed as having an unclear or high risk of bias for the majority of items on the Cochrane risk of bias tool (online supplementary file 2). Overall, the three non-randomised controlled studies were at risk of selection bias (online supplementary file 2). Ten of the 16 studies without a control group were prospective, while three were retrospective, and this aspect of the design was unclear for three. Overall, the 16 studies met between one and five of the nine quality criteria; only eight adequately described their inclusion criteria, and all were considered at risk of selection bias (online supplementary file 2). Poor reporting of study methods and results was a problem across all the study designs. Studies often made statements in the results section that were not backed up with numerical data. Where data were available, these were extracted; where the appropriate data were not reported, the description of the results provided in the paper was extracted, and the lack of summary data was noted in the data extraction tables, which are available in the HTA report.[1]

### Outcomes
The most systematically assessed outcomes in the included studies were gait quality and energy consumption, assessed during clinic/laboratory visits (table 2). While several studies (table 2) reported patient satisfaction with the device and functionality (eg, how it impacted sitting in their wheelchair; the main ways in which they used the device), the results were predominantly reported in an anecdotal fashion, and it was not possible to assess how robustly the information had been collected. Despite our requirement that participants in studies had used their orthoses outside the clinic, only one study used a validated measure (Barthel Index) of patients' ability to manage everyday activities of daily living outside the clinic setting[10]; and only two assessed quality of life using a validated measure (table 2). Generally, adverse effects such as skin damage or falls were not systematically reported. It cannot be inferred that there were few adverse events as authors did not specifically mention that no adverse events were identified.

### Patients with postpolio syndrome
Seven case series (n=143 patients) investigated types of carbon fibre KAFO (table 1). Three compared a new device to the one used previously by participants[11–13]; one compared using a device in stance control mode and locked mode (with the aim of replicating a traditional KAFO design)[14]; two before and after use of the orthosis[15 16]; and postintervention only in one study.[17]

Outcomes were sparsely reported. Five of the seven studies reported measures of patient satisfaction, although not in sufficient detail to assess the robustness of the evaluation.[11–13 16 17] Three studies made a formal assessment of walking ability,[11 12 14] and four assessed either energy consumption or particular muscle activity.[11 12 14 15] Resource utilisation data were limited to assessment of device malfunction in four studies[11–13 17] and cost in one study.[12] Five studies failed to report adverse effects data or to mention that no adverse effects were identified.[11 12 14–16]

### Inclusion body myositis
One case series study (n=9 patients) evaluated a stance control KAFO.[18] Gait was assessed in the clinic with and without use of the device following 6 months of use. A questionnaire was designed by the investigators to elicit patient outcomes, but the results were not reported in full. No data were reported on resource utilisation or adverse effects.

### Poststroke patients
Four studies (n=131 patients), one RCT,[19] a cohort study,[20] and two case series,[10 21] evaluated KAFOs and/or AFOs used for knee instability. One assessed a single outcome, gait with and without use of a carbon fibre KAFO.[21] Two studies compared a thermoplastic KAFO with an AFO for knee instability: one compared patients who had recovered sufficient control of knee activity to switch to an AFO with those who had not and therefore continued using a KAFO,[10] effectively the comparison was between those who had recovered sufficient control of knee activity to switch to an AFO compared with those who had not; the second compared KAFO with AFO and normal adult gait.[20] The RCT compared use of an AFO or KAFO with what was described as conventional rehabilitation (not reported in detail).[19] The only patient-reported outcome assessed was usage.[20] Three studies made a formal assessment of walking ability,[10 20 21] and two assessed other functional abilities.[10 19] None reported on resource utilisation data or adverse effects.

### Patients with spinal cord injury
Nine studies (n=194 patients), one RCT,[22] two controlled trials[23 24] and six case series,[25–30] evaluated HKAFOs. The RCT used a crossover design to compare Walkabout orthosis (WO) (Polymedic, Queensland, Australia) to an isocentric RGO (IRGO) (Center for Orthotics Design, Campbell, California, USA). There was a 2-month washout period of no orthoses use; the data were analysed as though from a parallel trial.[22] There were two further studies of WO with no comparator.[26 30] Two investigated a HGO, one with no comparator[28] and one compared the HGO with a custom-made RGO worn by the same patients in a crossover study.[24]

**Table 1** Study characteristics by condition

| Main publication (*associated papers*) Country | Study design N in study (n in analyses) | Population % male Mean age (SD) | Intervention (I) Comparator (C) | Cointerventions | Length of follow-up |
|---|---|---|---|---|---|
| *Postpolio syndrome* | | | | | |
| Bocker et al[15] (Bocker et al[36]) Germany | Case series n=10 (6) | 30 64.5 | I: carbon fibre KAFO C: no comparator | Gait training, pain therapy and exercises (twice per week for 3 months) | 3 months |
| Brehm et al[11] Netherlands | Case series n=23 (20) | 61 55 (9.2) | I: carbon fibre KAFO (locked knee-joint) C: leather/metal or plastic/metal KAFO used previously by same participants | Walking aids were used by some participants | 26 weeks |
| Davis et al[14] Australia | Case series n=10 (10) | 40 61.9 (7.7) | I: carbon fibre SCKAFO in stance control mode C: KAFO in locked knee mode used by same participants | Walking aids | Mean duration of use at time of evaluation 6.2 (SD 5.2) months |
| Hachisuka et al[12] Japan (Hachisuka et al[37]) | Case series n=11 (8–11*) | 18 53.9 (9.8) | I: carbon fibre KAFO C: traditional non-carbon KAFO used by same participants | Walking aids | Not reported |
| Heim et al[17] Israel | Case series n=30 (27) | 33 44 | I: carbon fibre KAFO C: no comparator | Not reported | 30 months |
| Peethambaran et al[13] USA | Case series n=5 (5) | 40 61.4 (12.4) | I: carbon titanium KAFO (anterior approach design) C: plastic KAFO (posterior approach design) used previously by the same participants | Not reported | 6 weeks |
| Steinfeldt et al[16] Germany | Case series n=55 (55) | 44 58 | I: carbon fibre KAFO C: no comparator | Not reported | >3 months |
| *Inclusion body myositis* | | | | | |
| Bernhardt et al[18] USA | Case series n=9 (6) | 78 61 (9) | I: SCKAFO C: no comparator | Not reported | 6 months |
| *Poststroke* | | | | | |
| Boudarham et al[21] France | Case series n=11 (unclear) | 64 51 (15) | I: carbon fibre KAFO C: no comparator | Not reported | Device prescribed within past 6 months |
| Kakurai and Akai[10] Japan | Case series n=28 (28) | 50 54.5 | I: plastic convertible KAFO (to AFO) C: participants who changed to AFO compared with those remaining on KAFO | Not reported | Not reported |
| Morinaka et al[20] Japan | Cohort study n=25 (25) | 64 56 | I: plastic KAFO C: 50 participants fitted with AFOs and a group of 30 healthy adult males | Not reported | Mean 14.6 months (range 1–35) |
| Yang et al[19] China | RCT n=67 (67) | 84 58 | I: KAFO or AFO C: 'Conventional rehabilitation' | Not reported | Not reported |
| *Spinal cord injury* | | | | | |
| Harvey et al[22] (Harvey et al[38], Harvey et al[39]) Australia | RCT (crossover) n=10 (5–10†) | 90 37 (8.4) | I: HKAFO (Walkabout orthosis) C: HKAFO (IRGO) | Gait training (30–54 hours per orthosis) Crutches | 28 weeks |

Continued

**Table 1** Continued

| Main publication (associated papers) Country | Study design N in study (n in analyses) | Population % male Mean age (SD) | Intervention (I) Comparator (C) | Cointerventions | Length of follow-up |
|---|---|---|---|---|---|
| Jaspers et al[25] Belgium | Case series n=14 (14) | 86 33.6 | I: HKAFO (ARGO) C: no comparator | Walker or crutches | 1 year |
| Middleton et al[26] Australia | Case series n=25 (21) | 76 35 (13) | I: HKAFO (Walkabout orthosis) C: no comparator | Parallel bars, forearm crutches or frames | ≥18 months |
| Scivoletto et al[27] Italy | Case series n=24 (24‡) | 79 33.6 (3.2) | I: HKAFO (RGO) C: no comparator (internal comparison of non-users vs users) | Not reported | 1 year |
| Summers et al[28] UK | Case series n=20 (20) | 100 28 | I: HKAFO (HGO ParaWalker) C: no comparator | Crutches used as decided by patient | Mean 20 months |
| Sun et al[29] China | Case series n=20 (15) | 67 33.7 | I: HKAFO (RGO) C: no comparator | Not reported | Not reported |
| Tang et al[23] China | Controlled study n=58 (unclear) | 83 32.4 | I: AGO, RGO, KAFO C: rehabilitation training | Rehabilitation training | 4 months§ |
| Whittle et al[24] UK | Controlled study (crossover) n=22 (Unclear¶) | 82 34 | I: HKAFO (HGO ParaWalker) C: HKAFO (RGO) | Rollator or crutches | 4 months |
| Wu et al[30] China | Case series n=6 (6) | 67 27.6 | I: HKAFO (Walkabout orthosis) C: no comparator group | Gait training including balance plus walking exercises | Unclear |

*Eight completed assessment of non-carbon fibre KAFO and walking without an orthosis and 11 completed assessment of carbon fibre KAFO.
†Appears to be 22 for analysis of final choice of orthosis, although one left the trial without trying either and three participants tried only one. It was unclear how many participants were included in other analyses.
‡Appears to be 10 for all analyses except for speed of walking on flat surface (n=8) and speed of walking on ramp (n=5).
§Eight weeks after fitting of device.
¶A total of 24 for the single outcome eligible for the review, although unclear for other analyses in the study.
AFO, ankle-foot orthosis; AGO, alternative gait orthosis; ARGO, advanced reciprocating gait orthosis; HGO, hip guidance orthosis; IRGO, isocentric reciprocating gait orthosis; KAFO, knee-ankle-foot orthosis; RGO, reciprocating gait orthosis; SCKAFO, stance-control knee-ankle-foot orthosis.

**Table 2** Outcomes assessed

| Study | Patient-reported outcomes | | | | Adverse effects | Objective assessments | | | Resource utilisation | |
|---|---|---|---|---|---|---|---|---|---|---|
| | Satisfaction with device | Functionality of device | Usage of device | Quality of life | | Walking ability | Energy consumption | Muscle activity | Device malfunction | Cost |
| **Postpolio** | | | | | | | | | | |
| Bocker et al[15] | | | | ✓ | | | | ✓ | | |
| Brehm et al[11] | ✓ | | | | | ✓ | ✓ | | ✓ | |
| Davis et al[14] | | | | | | ✓ | ✓ | | | |
| Hachisuka et al[12] | ✓ | | | | | ✓ | ✓ | | ✓ | ✓ |
| Heim et al[17] | ✓ | | ✓ | | ✓ | | | | ✓ | |
| Peethambaran[13] | ✓ | ✓ | | | ✓ | | | | | |
| Steinfeldt et al[16] | ✓ | ✓ | | | | | | | ✓ | |
| **Inclusion body myositis** | | | | | | | | | | |
| Bernhardt and Oh[18] | ✓ | ✓ | ✓ | | | ✓ | | | | |
| **Poststroke** | | | | | | | | | | |
| Boudarham et al[21] | | | | | | ✓ | | | | |
| Kakurai and Akai[10] | | | | | | ✓ | | ✓ | | |
| Morinaka et al[20] | | | ✓ | | | ✓ | | | | |
| Yang et al[19] | | | | | | | | ✓ | | |
| **Spinal cord injury** | | | | | | | | | | |
| Harvey et al[22] | ✓ | ✓ | ✓ | | | ✓ | ✓ | ✓ | | |
| Jaspers et al[25] | ✓ | ✓ | ✓ | | ✓ | | | | ✓ | |
| Middleton et al[26] | ✓ | ✓ | ✓ | | | | | | ✓ | |
| Scivoletto et al[27] | | ✓ | ✓ | | | | | | | |
| Summers et al[28] | ✓ | ✓ | ✓ | | ✓ | | | | ✓ | |
| Sun et al[29] | ✓ | ✓ | | | ✓ | ✓ | | | | |
| Tang et al[23] | ✓ | ✓ | | ✓ | | | | | | |
| Whittle et al[24] | ✓ | ✓ | | ✓ | | ✓ | | | ✓ | ✓ |
| Wu et al[30] | | ✓ | | | | ✓ | | ✓ | | |

The remaining four studies investigated types of RGO, two with no comparator,[25 29] one in comparison with RGO non-users[27] and one compared three different types of orthoses (plus rehabilitation training) with rehabilitation training alone.[23] Although each of the studies reported at least one patient-reported outcome, only one study reported using a validated scale (Barthel Index and Functional Independence Measure), and due to lack of clarity in the analysis and reporting, it is unclear whether there were any between-group differences at follow-up in this study.[23] There were fewer objective assessments across the studies than for the other conditions.[22 24 29 30] Resource utilisation data were limited to assessment of device malfunction[24–26 28] and cost.[24] Two-thirds of studies did not address adverse effects.[22–24 26 27 30]

## DISCUSSION
### Principal findings

The review identified a paucity of high-quality evidence assessing the effectiveness of orthotic devices for knee instability experienced by people with NMD and CNS conditions. In addition to the very limited use of robust study designs, in particular RCTs, reporting was generally poor. For example, several studies made statements about findings without presentation of supporting data. The evidence base consists of small, single-centre studies with outcome assessments that did not appear to have been undertaken independently of treating clinicians. Laboratory-based studies can provide useful insights about efficacy, particularly during development of a device. However, the literature is dominated by laboratory evaluations of orthoses: 76 studies were excluded because the evaluation of the orthosis did not include any use of the device by the patient in a non-clinic setting, and the most systematically assessed outcomes in the included studies focused on gait analysis and energy consumption. There was limited use of standardised, validated patient-reported outcome measures. In particular, there was an absence of evidence on outcomes that are reported by patients to be important to them such as reduction in pain, falls or trips, improved balance and stability and participation in paid employment, outdoor activities (such as gardening), family visits and social events.[1] In addition, fewer than one-third of the studies followed patients for a year or more. It is unlikely that studies of less than 1-year duration fully capture the effects of using the devices.

Given that patients report that orthotic devices prescribed for knee instability can play a crucial role in maintaining, promoting and enhancing physical and psychological health and well-being and participation in employment, family and social community activities,[1] the evidence gaps identified by our review are significant and important. A factor that might contribute to this discrepancy in outcome measurement is current requirements for device regulation; only evidence of performance and safety is required for medical devices associated with lower levels of risk to patients such as orthotics for knee instability. This may result in a lack of incentives to conduct primary research on efficacy and/or effectiveness.[31]

### Strengths and weaknesses of the study

We undertook systematic searches across an extensive range of sources for published, unpublished and ongoing studies. There were no language restrictions, and we included three studies published in Chinese[19 23 29] and one in German.[16] We assessed the risk of bias in the included studies and used standard methods to reduce error and bias at key stages of the review process. Several studies provided a descriptive report of some outcomes with no numerical data. Due to the paucity of evidence, we extracted these reports in order to provide as clear a picture as possible of what information is currently available. Arguably, this overestimates the amount of evidence that is available.

During study selection, it was often difficult to definitively determine whether the participants had knee instability. This was partly due to poor reporting and partly because knee instability was sometimes part of a more complex problem with stability and mobility and is not an explicit and well-defined clinical diagnosis. As a result, studies may have been included where it is arguable whether knee instability was the main problem and studies rejected that arguably do include people with knee instability. However, we would not expect that this would in any way change the overall conclusions of the review about the lack of high-quality evidence or allow conclusions to be made about the effectiveness of specific devices.

An evidence base of small single-centre studies and inadequate study design is similar to that identified in other reviews of orthotic devices for different populations.[32–34] Also a systematic review of questionnaires used to assess patient satisfaction with orthoses for any limb found that 63% of the 106 included papers used questionnaires developed for the specific study rather than validated measures, supporting our findings on this aspect of the evidence.[35]

### Unanswered questions and future research

There is a large gap in the evidence on the effectiveness of KAFOs, AFOs and other orthotic devices for managing knee instability related to NMD and CNS conditions. Robust research is required addressing outcomes that are important to patients. RCTs are the most robust way of assessing effectiveness, and a pragmatic trial that recognises that provision of an orthotic device is a complex intervention would be appropriate.

There are a number of challenges for researchers and clinicians to consider when designing future studies, including: defining the target population and knee instability, the personalisation of treatment including customisation of devices, the relative rarity of the problem within individual conditions and whether a trial including patients with knee instability with a range of NMD or CNS conditions would be generalisable. It may also be worth considering a national registry to systematically collect data on the ambulatory problem, devices provided, key elements of management of the instability, factors that inform/determine the

process of matching patients to orthotic devices, collection of a core set of standardised and validated patient-reported outcome measures, data on use of the device and resource use. While registries do have limitations, this would be a major step change from the current evidence base in terms of increased rigour and generalisability and would create a population database and an infrastructure from which future RCTs could be undertaken. The evidence base in this field could also be improved through systematic development of a core set of outcome measures (http://www.comet-initiative.org/). Future research regardless of study design should follow reporting standards (http://www.equator-network.org/).

## CONCLUSIONS

There is a need for high-quality research, in particular RCTs, on the effectiveness of KAFOs, AFOs and other orthotic devices for managing knee instability related to NMD and CNS conditions. This research should address outcomes that are important to patients. There may also be value in developing a national registry.

**Author affiliations**
[1]Department of Health Sciences, University of York, York, UK
[2]Centre for Reviews and Dissemination, University of York, York, UK
[3]Academic Unit of Health Economics, University of Leeds, Leeds, UK
[4]Department of Biomedical Engineering, University of Strathclyde, Glasgow, UK
[5]Queen Mary's Hospital, St George's University Hospitals NHS Foundation Trust, London, UK
[6]Leeds Institute of Rheumatic and Musculoskeletal Medicine, University of Leeds, Leeds, UK
[7]Royal Derby Hospital, Derby, UK
[8]Kingston University, Kingston-Upon-Thames, London, UK
[9]St George's University of London, London, UK

**Contributors** CM was responsible for writing the protocol and had overall responsibility for coordinating and leading the project, provided advice and input to all elements of the project and contributed to report writing. AB, DF, JO undertook study selection, quality assessment, report writing and contributed to the protocol. RR-L provided information specialist support, designed and undertook literature searches and wrote the related sections in the report. RB, CPI, SL, MP, GR, DM and RJO were members of the Advisory Group, contributed to the systematic review protocol and/or provided clinical and/or methodological advice throughout the review and commented on drafts of the systematic review report. CM drafted this manuscript, and all authors reviewed, edited and approved the manuscript.

**Funding** This project was funded by the National Institute for Health Research (NIHR) HTA Programme (project number 13/30/02) and has been published in full in Health Technol Assess 2016;20(55). Further information available at https://www.journalslibrary.nihr.ac.uk/programmes/hta/133002/#/This report presents independent research commissioned by the NIHR. The views and opinions expressed by authors in this publication are those of the authors and do not necessarily reflect those of the NHS, the NIHR, MRC, CCF, NETSCC, the HTA programme or the Department of Health.

**Competing interests** During this study, SL was an employee of Opcare, a company that provides orthotic and prosthetic services to the UK NHS. This company does not manufacture orthotic devices, although a sister company ORTHO C FAB does. CPI is a member of the National Institute for Health and Care Excellence Medical Technologies Assessment Committee and member of the European Clinical Research Infrastructure Network.

**Provenance and peer review** Not commissioned; externally peer reviewed.

**Data sharing statement** Most of the data are available in the main body and appendices of the HTA Monograph: https://www.journalslibrary.nihr.ac.uk/hta/hta20550/#/abstract. Any further data can be obtained from the corresponding author.

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
