## [Reviewer comments · BMJ Open]

ARTICLE DETAILS

TITLE (PROVISIONAL)	A SYSTEMATIC REVIEW OF THE EVIDENCE ON ORTHOTIC DEVICES FOR THE MANAGEMENT OF KNEE INSTABILITY RELATED TO NEUROMUSCULAR AND CENTRAL NERVOUS SYSTEM DISORDERS
AUTHORS	McDaid, Catriona; Fayer, Debra; Booth, Alison; O'Connor, Joanne; Rodriguez-Lopez, Rocio; Mccaughan, Dorothy; Bowers, RJ; Iglesias, C; Lator, Simon; O'Connor, Rory J.; Phillips, Margaret Philip; Ramdharry, Gita

VERSION 1 - REVIEW

REVIEWER	Ayeni, Olufemi McMaster University, Canada
REVIEW RETURNED	19-Jan-2017

GENERAL COMMENTS	Dear Author, good paper with some concerns. 1) search is over 2 years old and needs upgrading to 2017 2) How were disagreements resolved between reviewers, either document exactly how (senior author, consensus meeting) or report kappa agreement statistics! 3) Conclusion needs strengthening, as the message that research needs to improve is all too common, focus on the "signals" found in the results, what worked and what did not and for which conditions despite the limitations of the quality of research 4) Both intro and discussion can be condensed.
---

REVIEWER	Roshanak Baghaei Roodsari University of social welfare and rehabilitation science
REVIEW RETURNED	03-Apr-2017

GENERAL COMMENTS	I accept this article
-----------------------

REVIEWER	Sarah Tyson University of Manchester UK
REVIEW RETURNED	13-Jun-2017

GENERAL COMMENTS	This is a very well structured, methodologically sound, beautifully presented review. I have found no points that require clarification or revision. This is only the second time in my career I have recommended acceptance without any revisions. Congratulations
---

REVIEWER	Éder Kröeff Cardoso Centro Universitário Cenecista de Osório - UNICNEC Brazil
REVIEW RETURNED	21-Jun-2017

GENERAL COMMENTS	Research relevant to the clinical and scientific environment. Some considerations could be included: 1. Insert in the methodology the research problem that justified the literary search. 2. Clearly state the exclusion criteria of articles
--

VERSION 1 – AUTHOR RESPONSE

Reviewer 1

1) search is over 2 years old and needs upgrading to 2017

Response: We understand this valid concern about the searches. A review is potentially out of date from the day after the searches have been completed – however this is entirely dependent upon what has been published in the meantime. A review effectively only becomes out of date when there has been research published which may change the findings and conclusions of the review: this is one of the factors in prioritising whether a review is updated. Given the lack of RCTs in this field and the poor quality and limited scope of the three registered ongoing studies that we identified as part of the review it is highly unlikely that any research has been published in the meantime that would change the conclusions of the review. This was a very comprehensive review with multiple databases and other sources searched without language restrictions and any update is a substantial undertaking and a wasteful use of research resources if an update is not justified, especially given that funding for the project has finished.

2) How were disagreements resolved between reviewers, either document exactly how (senior author, consensus meeting) or report kappa agreement statistics!

Response: Disagreements were resolved through discussion and consultation with a third member of the project team if necessary. This information is provided on page 6.

3) Conclusion needs strengthening, as the message that research needs to improve is all too common, focus on the "signals" found in the results, what worked and what did not and for which conditions despite the limitations of the quality of research

Response: We agree that the conclusion that the research needs to improve is too common, but unfortunately this very much reflects what we found. The majority of studies were very poorly reported and poorly designed and in our view there are no signals for specific devices.

4) Both intro and discussion can be condensed.

Response: We are happy to undertake condensing these sections of the paper – could we please have editorial guidance on how much the word count needs reduced by.

Reviewer 2

No amendments required

Reviewer 3

This is a very well structured, methodologically sound, beautifully presented review. I have found no

points that require clarification or revision. This is only the second time in my career I have recommended acceptance without any revisions. Congratulations

Response: Thank-you!

Reviewer 4

Research relevant to the clinical and scientific environment. Some considerations could be included:

1. Insert in the methodology the research problem that justified the literary search.

Response: This has been added to Methods section on P 4.

2. Clearly state the exclusion criteria of articles

Response: The exclusion criteria were the reverse of our inclusion criteria. These have been added where relevant.